# Mask R-CNN Based C. Elegans Detection with a DIY Microscope

**DOI:** 10.3390/bios11080257

**Published:** 2021-07-30

**Authors:** Sebastian Fudickar, Eike Jannik Nustede, Eike Dreyer, Julia Bornhorst

**Affiliations:** 1Assistance Systems and Medical Device Technology, Faculty of Medicine and Health Sciences, CvO University of Oldenburg, Ammerländer Heerstraße 140, 26129 Oldenburg, Germany; eike.jannik.nustede@uni-oldenburg.de (E.J.N.); eike.dreyer@uni-oldenburg.de (E.D.); 2Faculty Mathematik und Naturwissenschaften, Bergische Universität Wuppertal, Lebensmittelchemie, Gaußstr. 20, 42119 Wuppertal, Germany; bornhorst@uni-wuppertal.de

**Keywords:** C. elegans, segmentation, classification, DIY microscope, mask R-CNN

## Abstract

Caenorhabditis elegans (C. elegans) is an important model organism for studying molecular genetics, developmental biology, neuroscience, and cell biology. Advantages of the model organism include its rapid development and aging, easy cultivation, and genetic tractability. C. elegans has been proven to be a well-suited model to study toxicity with identified toxic compounds closely matching those observed in mammals. For phenotypic screening, especially the worm number and the locomotion are of central importance. Traditional methods such as human counting or analyzing high-resolution microscope images are time-consuming and rather low throughput. The article explores the feasibility of low-cost, low-resolution do-it-yourself microscopes for image acquisition and automated evaluation by deep learning methods to reduce cost and allow high-throughput screening strategies. An image acquisition system is proposed within these constraints and used to create a large data-set of whole Petri dishes containing C. elegans. By utilizing the object detection framework Mask R-CNN, the nematodes are located, classified, and their contours predicted. The system has a precision of 0.96 and a recall of 0.956, resulting in an F1-Score of 0.958. Considering only correctly located C. elegans with an AP@0.5 IoU, the system achieved an average precision of 0.902 and a corresponding F1 Score of 0.906.

## 1. Introduction

Studies about diseases and toxicity are important to evaluate the safety of chemicals, pharmaceuticals, and identify toxic properties of food constituents. The welfare of animals used in research is essential and guiding principles nowadays, including replacement, reduction, and refinement (3 R principle) of animal experiments. Using the nematode Caenorhabditis elegans (C. elegans) helps to reduce the number of animal testing drastically by providing the option of high-throughput toxicity screening. Advantages such as their high procreation rate, uncomplicated, cheap maintenance, and short life-cycle provide ample entry points for various studies. Morphological changes of the worm can easily be seen under a microscope, as C. elegans are transparent in brightfield images. In multiple studies, C. elegans has been proven to be a well-suited model to study toxicity, and the identified compounds were closely matching to compounds identified as toxic in mammals [1]. A benefit of the alternative animal model compared to in vitro models include its conserved drug metabolism, which is remarkably similar between C. elegans and mammals with phase I and II enzyme modifications to drug compounds [2].

For standardized quantification and characterization, various computer vision approaches have been confirmed to apply for the automatic detection of worms [3,4,5]. Existing approaches rely on brightfield microscope images and apply classic computer vision algorithms. Worms are separated via edge detection [6], and individual features such as head-to-tail thickness ratios are calculated [7]. A robust segmentation algorithm for C. elegans in agar was developed by Kabra et al. [8] using high-resolution images of single worms (1388 × 1036 px) to segment worms from their respective tracks in agar. The WormSizer plugin [9] of the Fiji ImageJ implementation [10] determines the growth and shape of mutant worms. While achieving a promising detection rate, a low sensitivity was found for overlapping and intersecting worms, which correspondingly could not be analyzed. The Matlab implementation CeleST [11] can detect 4–5 adults per plate. The WorMachine framework [12] uses Matlab-based machine learning algorithms for analysis. It includes a convolutional neural network (CNN) for worm identification and filtering of faulty worms (defined as overlapping or intersecting worms) based on high-resolution brightfield images acquired with an Olympus IX83 microscope. The worms are identified and cropped via Matlab’s Image Processing Toolbox. The cropped worms (64 × 128 px) are then classified into valid or faulty worms via a CNN. Features of interest were extracted from valid worms and used as inputs for machine learning algorithms. For example, WorMachine uses a Support Vector Machine (SVM) for binary classification of the worm gender. The CNN was trained on 11.820 binary masks of worms, with 85% of images used for training and 15% for testing and 99.2% accuracy on the training set, and 93.4% in the test set. With the previous approaches requiring high resolution (microscopy) images and focusing on tracking single worms at a time, instead of multiple, the discussed approaches still require a manual preselection of C. elegans and the availability of a high-resolution microscope. Consequently, analysis and evaluation of corresponding C. elegans studies are slowed by tedious manual evaluations, i.e., manually counting and characterizing the nematodes one by one under a microscope. This requires time and a constant level of focus to reduce intra-test and inter-tester variability to assure comparability among studies and researchers. As a result, study sizes are currently mainly constrained by the number of researchers and microscopes.

To alleviate these problems, we proposed an approach [13] that utilizes low-cost smartphones within a do-it-yourself microscope setup to capture complete Petri dishes and identify multiple C. elegans via a SVM [14]. This approach enabled the capturing of complete Petri dishes with nematodes with a recall of 0.90 and specificity of 0.85 via the included illumination, optics, and the smartphone camera. However, the previous approach held these limitations: By applying a classic machine learning algorithm with handcrafted features, the resulting high memory, and computational complexity limited the step size of the sliding window for feature extraction with on-smartphone processing to a 50% overlap. Another concern was the variations of focal lengths and firmware (applying various pre-processing steps and autofocus methods) among mobile phone cameras, resulting in the necessity of readjustment per camera model.

The practicability of specialized auto-encoders, for instance, segmentation to detect and follow worms, has been shown by Javer et al. [15] for images with 5 to 10 worms. They utilized different features for classification, including eigenworms [16] and skeleton angles, represented as embedded time vectors and indicated a high sensitivity of skeleton angles to identify mutant worms with different strains via their posture, which the angles of the worm skeleton may represent the best. Akintayo et al. [17] proposed the combined use of region proposals and auto-encoder network to detect and count nematode eggs. A sliding window was applied on a microscopic image. The frame was then filtered before being fed into the fully convolutional network (FCN). By applying small steps for the sliding window and only considering frames that contained whole eggs (no cut-offs at the edges), the approach is very similar to a region proposal approach. Instance labels are assigned to each egg via a matrix label function considering the segmentation map of eggs to the background generated by the FCN.

Considering these limitations, the herein proposed approach enhances the concept of the previous system in following two directions: Aiming for a generalized system to guarantee standardized image acquisition conditions, we propose a low-cost do it yourself (DIY) microscope with a fixed camera and a controlling unit. In addition, to improve the detection accuracy, we shift the algorithmic approach from object detection to instance segmentation, which might enable a further qualitative analysis of worm locomotion and health assessment by movement and size. Thereby, we replace the SVM-based classification approach with a Mask R-CNN (MRCNN) object detector.

## 2. Methods

### 2.1. Optical and Processing System

The previously proposed smartphone based C. elegans tracker [13] encountered significant issues, including memory and compute-power constraints of the smartphone. With the chosen sliding window approach and a 50% overlap between subsequent windows, a reduced image resolution of 3024 × 4032 px was shown to be maximally practical for smartphones’ processing capabilities, while a higher image resolution was expected to result in higher accuracy. Cut worms among window borders were skipped entirely.

Moreover, camera model variations require adjustments (e.g., with varying focal lengths, distance adjustments between lens and Petri dish become necessary) and camera-specific training of classifiers. Due to the focal length variations, capturing whole agar plates was only possible with selected camera models. To overcome the variability in camera, magnification, capture distance, and lighting, we developed a low-cost hardware setup with an integrated optical and processing unit instead of the previous smartphone-based solution. With the herein proposed hardware setup, stable recording conditions can be assured, and the classifier does not require further adjustments, such as retraining.

Our new version in Figure 1, a Raspberry Pi Model 3B+, in conjunction with an 8MP Raspberry Pi Camera Module V2.1, is used for image acquisition. A 2 × zoom telephoto objective lens is combined with a 14 × macro lens for optimal magnification of the agar plate onto the camera sensor for the optical part. The zoom lens magnifies the agar plate while enabling an increased distance to capture whole Petri dishes (3.5 cm). The macro lens further increases the magnification and supports focusing much closer objects, decreasing the distance between the Petri dish and the sensor. During image capturing, any background light is excluded by closing the recording unit. With these components, an even higher cost-efficiency is achieved since the necessity of a smartphone is omitted, and the overall material costs of this system range around EUR 200.

JPEG images were captured at a frequency of 1 Hz at 3280 × 2464 pixels as the maximum reliable recording frequency for the Raspberry Pi. A still image capture mode was used to overcome the limited image dimensions of 1920 × 1080 pixels for video recordings on the Raspberry Pi. The image acquisition is implemented in Python. Each image was stored with a unique name, date, time, and start time to differentiate between image sequences. The camera was controlled by the pi-camera Python module version 1.13. With the automatic camera attribute adjustments of brightness, exposure mode, and shutter speed, resulting in temporal aliasing effects due to the interaction between led frequency and automatic shutter speed, backlight illumination mode and a shutter speed of 25 s were configured, as determined best in testing suitable exposure modes (backlight, very long, fixedfps) at varying shutter speeds (15, 20, 25, 30, 35, and 40 s). Exposure time is, by default, set and limited by the frame rate of the cameras background video stream. With a maximum frame rate of 15 fps at stated resolution, corresponding maximum exposure time is fixed at $66\frac{2}{3}$ ms. Finally, these parameters were chosen according to how well we, as untrained observers, were able to identify worms on the captured images. Unmentioned parameters were left at the default settings for backlight illumination. A resulting unprocessed JPEG image is shown in Figure 2. The dataset is available via https://doi.org/10.5281/zenodo.5144802 (accessed on 26 July 2021).

The microscope supports connectivity via both WiFi or USB and can be operated by either computers or mobile phones, e.g., via a web-based interface running on the Pi itself.

### 2.2. Pre-Processing Steps

As pre-processing steps, edge-region removal and sliding-window have been applied to the captured images. Since the edge regions of Petri dishes are particularly contrast-rich and frequently hold false-positively detected C. elegans and do not hold any C. elegans, these edge regions have been removed from the image in a pre-processing step. Therefore, vertical edges of the Petri dish were initially detected by scanning images’ horizontal center column for characteristic variations in luminance values. Based on the detected vertical edges, the diameter and center of the Petri dish were determined, and all pixels outside of the resulting circle are colored blacked, as shown in Figure 3b.

Images are divided into suitable chunks via a sliding window approach to enable image processing via the graphic card and parallel processing. In contrast to the previous approach, we use the full image resolution of 3280 × 2464 pixels. A chunk size of 820 × 821 pixels, including an overlap of 25%, was applied. The reference masks (for training and evaluation C. elegans) were prepared correspondingly—with one mask encoding a single worm.

### 2.3. ML-Based Segmentation

For nematode detection, a contour detection approach is used with low computational complexity and memory load while simultaneously increasing detection accuracy, especially of C. elegans close to each other, and enables the calculation of their size and precise location. Thereby, we utilize the effect that fully convolutional networks can be arranged as auto-encoders. The initial encoder stage uses convolutional layers with a stride >1 to reduce the spatial dimension until the input is reduced to a linear representation. The decoder then generates a segmentation map by up-sampling the linear features to the same dimension as the input with deconvolution layers.

We used a Mask R-CNN (MRCNN) [18] object detector framework to detect worm positions accurately. Since the region proposal network reduces the to-be classified areas and regulates the proposed regions, the largest anchor box that fits the object is always chosen, and a significant reduction in computational-complexity is achieved. The full convolution network of the mask branch also helps to identify multiple worms in one anchor box as it classifies pixel-by-pixel. We preferred the region-based method over the regression/classification networks you only look once (YOLO) [19] and Single Shot MultiBox Detector (SSD) [20] for the following reason: Even though SSD and YOLO detectors have experienced great success, especially on smartphones and embedded systems due to their small size and fast computation, they have been found to not work well on large images with small objects [21] when compared to region proposal approaches.

The Mask R-CNN implementation of [22] written in Python with the packages Keras (version 2.3.1) and Tensorflow (GPU version 1.13.2) (shown in Figure 4) was adapted for worm recognition. This implementation also supplied tools to investigate the trained models and data-sets besides being highly customizable. We adjusted the anchor box size, the number of region proposals, ground truths, and the maximum detection per image by visual inspection of samples of the data-set and generating bounding boxes based on extracted masks. Non-maxima suppression (NMS) [23] was used to filter region proposals with an overlap ratio of at least 0.5 in a proposal layer, added after the RPN. Corresponding overlapping regions were suppressed, as they may be redundant, i.e., showing the same object.

The inference model included an additional layer, the detection layer, to filter our predictions. For this purpose, the classifier head of the inference model did not run in parallel as it did during training but in sequence instead. The RPN was limited to a fixed maximum number of region proposals (with an RPN anchor scale of 18) after NMS. After the proposal network, the classifier head first generated the class labels with associated confidence scores and bounding boxes. Afterward, a threshold was applied to the confidence of class labels.

All predicted boxes with background labels were removed since they do not concern us and would only clutter the image. Per-class NMS was applied to remove redundant boxes and determine the most accurate representation of each instance. A fixed number of top-scoring boxes (class scores) had their mask predicted by the mask branch. This approach limits the mask computation to the most accurate boxes (with a network confidence level of at least 0.92) and is the preferred way for inference by the original authors of Mask R-CNN.

After bounding-box prediction, all predicted boxes that covered less than 700 square pixels and with an edge-length below 24 pixels were removed in a post-processing filter-step, being found too small for meaningful segments.

### 2.4. Study Design

For training of the Mask R-CNN model and evaluation of the overall nematode detection system, the following study design was chosen:

Recording: Within the defined optical and processing System, 1908 images were taken with 3 Petri dishes in the following manner: Each acquisition covered 5 min (300 images). Between acquisitions, at least 30 min passed to ensure significant movement of worms to interpret individual acquisition sequences as new Petri dishes.

Annotation: C. elegans in all images have been initially annotated via the data pre-processing steps discussed in Section 2.4.1. Subsequently, the resulting masks and images have been visually inspected and adjusted where necessary. Annotators were instructed that labels should represent minimal bounding boxes of each C. elegans. An example for annotations is shown in Figure 5.

Image masks of 768 × 768 pixels were used as representing the most suitable warp-size of Mask R-CNN. The data-set was split into 1345 images for training (70%), 366 testing (19%), and 197 validation (10%) set.

#### 2.4.1. Data Pre-Processing

The following pre-processing steps were used from the OpenCV 2.4. library (as shown in Figure 6) for creating and labeling the data-set, but are not required during detection.

Bilateral filtering was used to denoise the image by smoothing while preserving sharp edges by representing each pixel value by a weighted average of neighboring pixels with a distance between pixels of 9, a color distribution of 10, and a space of 5.Afterward, images are converted to grayscale.Image binarization was done by Gaussian adaptive thresholding—where the difference chooses the threshold pixel value between the weighted mean pixel value of a 71 pixel window and a constant of 4.Morphological operations were used to filter out small, unwanted dots usually present in the images at this stage. First, a closing operation, a dilatation followed by erosion, is applied to close holes in larger objects. Afterward, the opening operation, an erosion followed by dilatation, removes white dots [24].Larger, blob-like objects such as trapped air or gases in the agar were filtered via a Blob detection process. Thereby, the contours of each object are determined and filtered according to their area (from 0 to 700), convexity (from 0.3 to 1), circularity (over 0.2), and inertia (ratio from 0.2 to 1).To exclude small segments, masks that bounding boxes covered less than 700 square pixels and an edge-length below 24 pixels were removed according to the post-processing filter-step.Mask extraction was carried out by another contour filter application. This time, contours similar to worms in terms of convexity and inertia but differ in the area from the average adult worm were filtered out. This was a semi-automatic process as we had to assess whether the contour chosen as a mask represented a worm or not.

The resulting masks were used as a reference during the training and evaluation of the model.

#### 2.4.2. Model Training

For model training, an Nvidia Tesla P100- PCIE-16GB GPU with a batch size of 2 images was used. Each image was limited to 256 sampled regions of interest, with a ratio of 1:3 of worms to the background. Due to its better performance on many challenges when compared to, e.g., VGG16 and ResNet-50, we chose the ResNet-101 as our backbone network. For the region proposal network, five square anchor box lengths (32, 64, 86, 128, and 172 pixels) were chosen with width-to-height ratios of 0.5, 1, and 2. Lengths were inspired by our previous work and inspecting our data-set by generating bounding boxes from extracted masks. The ratios are supposed to capture all kinds of worm orientations while being as compact as possible per worm. The NMS threshold was set to 0.7 to ensure enough boxes around worms as they might be close to one another. Other parameters were set following [18,25] as they are more of a general nature and were found to be robust for most tasks.

A learning rate of 0.001, the momentum of 0.9, and weight decay of 0.0001 were used for training. Finally, all layers (backbone, FPN, RPN, and the three classifier heads) were trained on over 100 epochs to create the most accurate model. Moreover, to enhance the model’s robustness towards erroneously detecting C. elegans in the background region, the optimal model has been trained again for 60 epochs on the same data-set without applying background removal.

The validation-set was used for logging purposes (loss on validation set). For inference, we limited the total proposals to 1000 and the maximum number of detections to the 100 top confidence scores as in [18]. Conveniently, the latter suited our data-set, and the maximum number of ground truths per image was 53. The minimum detection confidence was set to 0 since only two classes (worm or background) were considered. Hence filtering by minimum confidence is not required as the minimum confidence will be at least 0.5. The NMS threshold was set to 0.3 for per-class NMS, which is the same as in [18]. The proficiency of the resulting model was evaluated by matching ground truths to prediction boxes and masks.

#### 2.4.3. Evaluation Metric

As the Receiver Operating Characteristics (ROC) curves are generally not applicable to object detectors due to the consideration of true negatives, which, in the case of object detectors, represent background, the average precision (AP) at 0.5 intersection over union (IoU) has been used for evaluation. These metrics represent the current de facto standard for evaluating object detectors.

The average precision (AP) value is defined as the area under the precision-recall curve (PRC), as proposed by the Pascal VOC Challenge [26]. The curve is sampled only at recall values for the AP calculation, where the maximum precision value drops.

While matching our predictions to ground truths, the prediction had to have at least an IoU of 0.5 with the tested ground truth to be a match. Ground truths without matches were counted as false negatives, predictions without matches as false positives.

## 3. Evaluation

Within the 1908 captured images, a total of 52,657 worm masks have been extracted with our pre-processing method, including visual inspection: 36.906 masks in training, 10.255 masks in testing, and 5496 masks in the validation data-set.

We evaluate the model’s ability to detect nematodes by considering the IoU among the annotated and detected bounding boxes. Table 1 indicate that nematodes have been detected with an F1 Score of around 0.96, and both recall and precision were well balanced. Besides the general detection of nematodes, accurate segmentation is essential for calculating the nematodes’ length and movement-rate. Thus, we also evaluated the model’s sensitivity regarding its IoU by applying a threshold of 0.5 for the IoU. Considering the classification accuracy in relation to minimal acceptable IoU (see Figure 7 we found IoU of 0.5 representing the most suitable trade-off among a sufficient high IoU and a suitable classification accuracy. The model achieved a corresponding F1-Score of around 0.9 (see Table 1—column “AP@0.5 IoU”).

A resulting visualization of the predictions for a Petri dish is shown in Figure 8, and a corresponding annotation and prediction are shown in Figure 9a. Each worm instance is visualized by its predicted mask (blue) and its bounding box (red) and gets assigned an ID for further analysis.

## 4. Discussion

Based on the general aim to develop a do-it-yourself microscope setup to capture complete Petri dishes and identify multiple C. elegans, we faced challenges with our previous smartphone-based system [13], such as variations of focal lengths and firmware (applying various pre-processing steps and autofocus methods) among mobile phone-cameras and computational-complexity of the applied Support Vector Machine. Thus, in this article, we propose a low-cost DIY microscope with a fixed camera and a controlling unit (based on a Raspberry Pi) to guarantee standardized image acquisition conditions and overcome the necessity of readjustment per camera model. This alternation of hardware enabled the system to gather 3280 × 2464 pixel images at 1 Hz and process them in 820 × 821 pixel chunks. Moreover, standardized optics and lightning conditions overcome the necessity to readjust the machine learning models per camera model.

We integrated a combined region proposal and specialized auto-encoder as a Mask R-CNN (MRCNN) object detector of C. elegans to enhance detection accuracy and lower the computational-complexity and memory-load associated with classic SVM algorithms. In addition, to improve detection accuracy, we shift the algorithmic approach from object detection to instance segmentation, which might enable further qualitative analysis of worm locomotion and health assessment by movement and size.

Considering the object detection results (as compared via the IoU of the bounding boxes), we found a high accuracy—as indicated by an F1 Score of 0.96 with correspondingly high precision and recall. Thus, we found an accuracy gain, compared to the previous HW and SW setup that had a recall of 0.90 and a specificity of 0.85 [14]. However, since both systems have been evaluated with different Petri dishes, the comparability among both systems might be only partially given, even though a representative data-set has been used in both cases.

Our results are remarkably similar to the accuracy of the approach of [15], which achieved an F1 Score of 0.98 in detecting C. elegans via skeletons angles and eigenworms on the part of the divergent set from the C. elegans Natural Diversity Resource (CeNDR) [16,27]. However, the latter evaluation used high resolution microscope images instead of standard consumer-grade resolution cameras as sensors. As this reference evaluation considered 5 to 10 worms per Petri dish, 24.47 C. elegans were covered in average per single dish in our setting.

Considering the ability to segment the worms, we found a lower F1-score of 0.906 if applying a minimal IoU of 0.5. Considering this reduced accuracy compared to the bounding-box based detection might suggest that a general loss of precision is experienced to detect nematode-specific contours. This might indicate a potential limit for the recording conditions (including lighting, Petri dish conditions, and camera resolution). However, it must also be considered that due to the limited image resolution, errors might occur in both the calculated bounding boxes and the manually labeled bounding boxes. In particular, in manually detected C. elegans and correspondingly segmented nematodes, the masks were regularly slightly larger, potentially resulting in a false negative miss due to an IoU over 0.5 (see Figure 9a. For example, the nematode ID 4 in Figure 9b the resulting overlapping pixel among the annotation and the detected segment are 211 while allover 364 pixels are covered, thus resulting in a IoU of 0.58. For this example, even though the nematode is well captured, it would have nearly been considered as False Positive.

We can assume some C. elegans were excluded due to their small size; even though we knew the challenge to specify to which degree nematodes should be included and addressed this need via a guideline and an additional automatic post filtering in both the manual annotations and the evaluation results, some corresponding artifacts may have remained. As an example, we show in Figure 10 the case of an enrolled nematode, which has not been annotated but was still detected by the automatic system.

Such nematodes that were falsely deemed false positives exhibited rather high confidence scores. To test this hypothesis, we looked at more sample images. Taking the image shown in Figure 10 as another example, we found that the false positives were worms.

In addition, even some false negative annotations have been found on visual inspection (see, e.g., Figure 11).

Visual inspection of the results shows weaknesses in the classification. For example, occasionally, worms that can be seen very well and are not visibly different from the precisely detected worms are not segmented. Inaccuracies also occur in segmentation, especially when C. elegans are in vesicles. With worms crossing, these will usually be recognized as one worm (see Figure 12a. Figure 12b represents another example, where a worm’s mask is faultily classified as the worm passes through a blob.

However, based on our visual inspection, we recognized most of these errors do not persist over multiple subsequent images, and thus might be overcome once considering sequential images in a time-series. Thus, we are confident that the resulting artifacts will not hold relevant for the intended use while being remarkable for the given study. However, the mislabeled worms are hard to see with the human eye as they are quite blurry and/or small. It is exceptional that the detector can detect these, but why were they missed during annotation? We applied semi-automatic mask extraction; using image processing to initially select masks. We assume that they must have been filtered out usually and expected these barely visible worms to be missed in manual inspection since they are either very small, blurry, or in areas with large nutrition blocks (darker than the rest). Hence it made sense to exclude them during mask extraction as we were unsure whether they were worms. With this in mind, we can assume our detector performs significantly better than expected. The given statistics, or rather the precision values, cannot represent the current models accurately.

## 5. Conclusions

In this article, a low cost system is presented consisting of a low-cost DIY microscope with a fixed camera and a controlling unit (based on a Raspberry Pi) to guarantee standardized image acquisition conditions. By utilizing the object detection framework Mask R-CNN, the nematodes are located, classified, and their contours are predicted with high precision. One advantage of this setup is the detection of the worms in an agar-filled Petri dish. Worms can be treated with a toxin first and put on a dish afterwards, or the toxic compound can be added to the E.coli (worms food source). The worm number detected by the system allows identifying a toxic potential of the compound. The advantage of our system is that it is easy to handle since you just need to put the Petri dish in the “holder”. Therefore, the worms are not additionally stressed, and you can easily image the plate again to observe the time and concentration dependency of the toxin. The system supports high-throughput toxicity screening in an alive and metabolizing organism. In the next step, we will investigate the system’s suitability for tracking nematodes over image sequences by investigating the ability to determine further phenotypic worm parameters. Overall, we developed a low-cost, end-to-end automated detection system for C. elegans that yields reliable, precise information of the nematode location affordable for various laboratories in developing and industrial countries.

## Figures and Tables

**Figure 1 biosensors-11-00257-f001:**
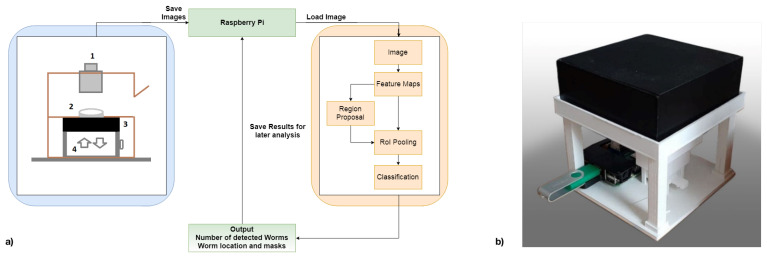
(**a**) Schematic of the whole prototype system structure. The hardware, on the left, consists of (1) the camera module with lenses, (2) the agar plate with C. elegans, (3) the lighting system, reused from [13] and (4) a movable stage to determine the correct distance to the lenses and thus guarantee sharp images (only in this prototype system, distance will be fixed later). The Pi saves taken images on itself. Loading an image into the detector yields the number of detected worms and corresponding bounding boxes and masks, which can be drawn onto the original image or saved separately. (**b**) A subsequent version of the DIY microscope, where the optics and lighting has been flipped vertically.

**Figure 2 biosensors-11-00257-f002:**
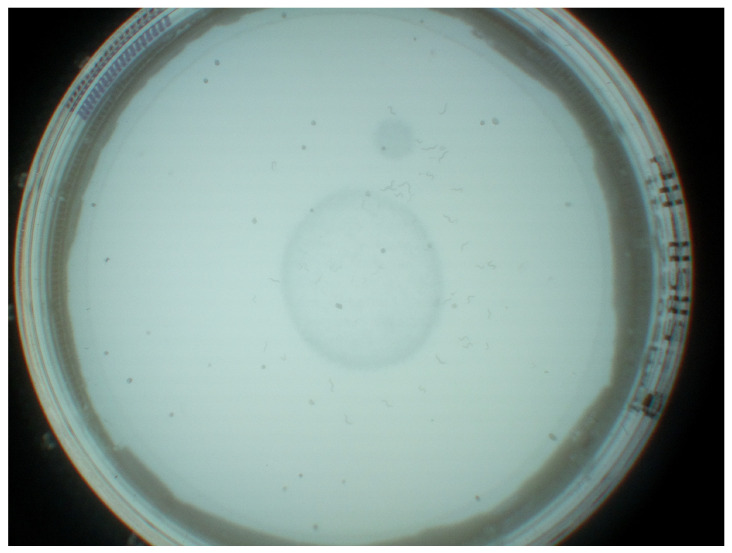
Example unprocessed JPEG image with pre-processing applied.

**Figure 3 biosensors-11-00257-f003:**
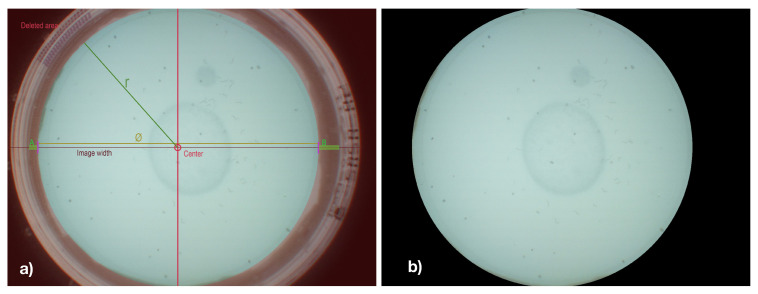
The surrounding pixel and the edge region of the Petri dish are removed from consideration (by blackening them out). Figure (**a**) Showing the original image and highlighting relevant information, (**b**) showing the resulting processed image).

**Figure 4 biosensors-11-00257-f004:**
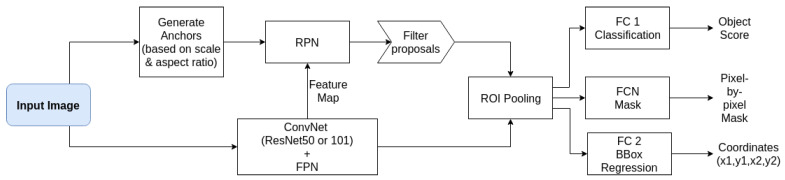
Workflow diagram of the applied Faster R-CNN network.

**Figure 5 biosensors-11-00257-f005:**
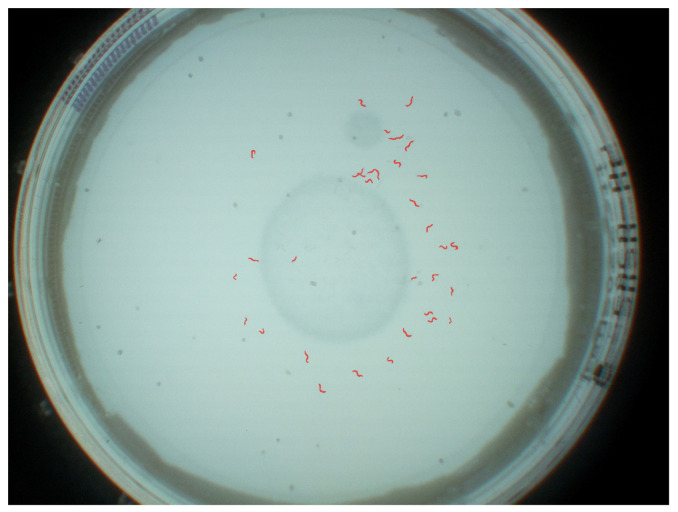
An example image with annotated segments of C. elegans.

**Figure 6 biosensors-11-00257-f006:**

Image pre-processing steps for the generation of reference masks (as annotations).

**Figure 7 biosensors-11-00257-f007:**
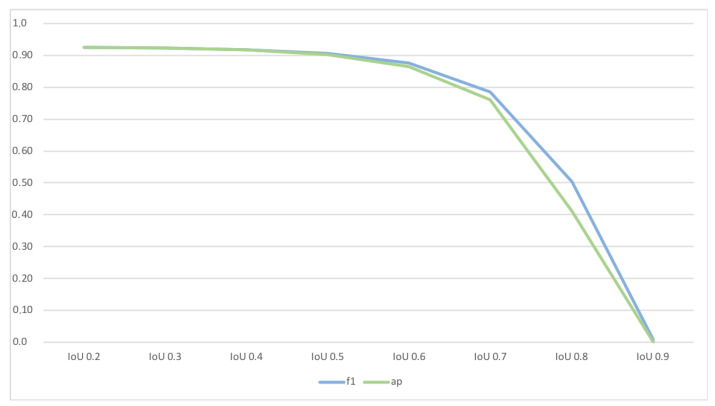
Classification accuracy in relation to minimal acceptable IoU—representing degree of IoU among nematode samples.

**Figure 8 biosensors-11-00257-f008:**
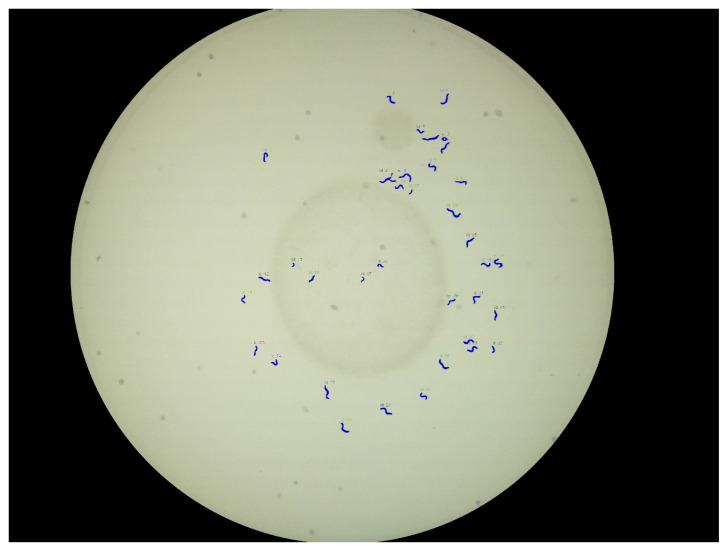
Resulting classified image-version of the example image: blue masks result from model-classifier; red and yellow: bounding boxes, nematode-IDs.

**Figure 9 biosensors-11-00257-f009:**
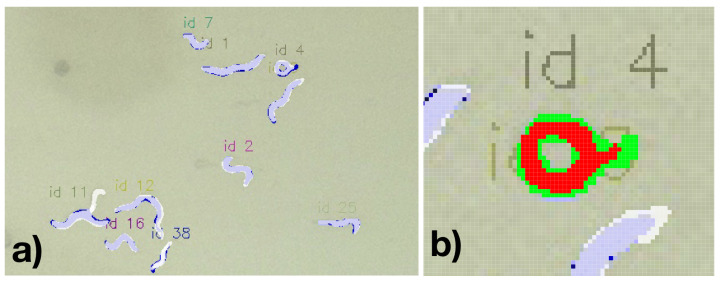
(**a**) Detected (blue) and annotated (white) nematode mask and (**b**) for a specific nematode (overlapping pixels among annotation and detected red).

**Figure 10 biosensors-11-00257-f010:**
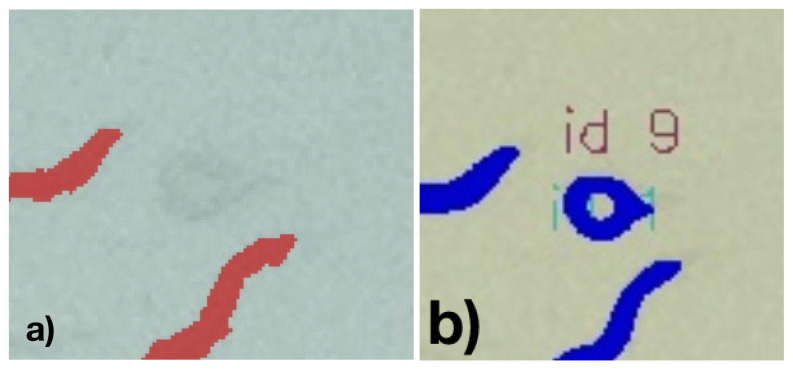
False positively detected nematode—due to a missing annotation: (**a**) annotated nematode image and (**b**) segmentation result.

**Figure 11 biosensors-11-00257-f011:**
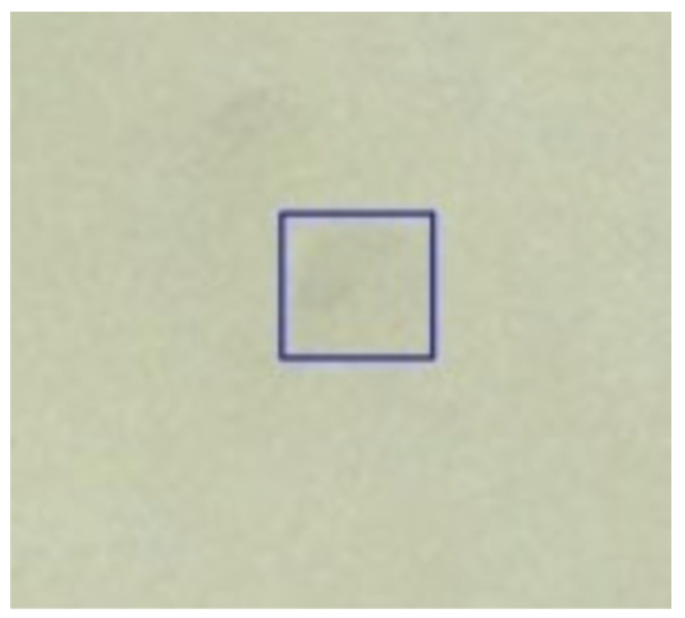
False negative detected nematode.

**Figure 12 biosensors-11-00257-f012:**
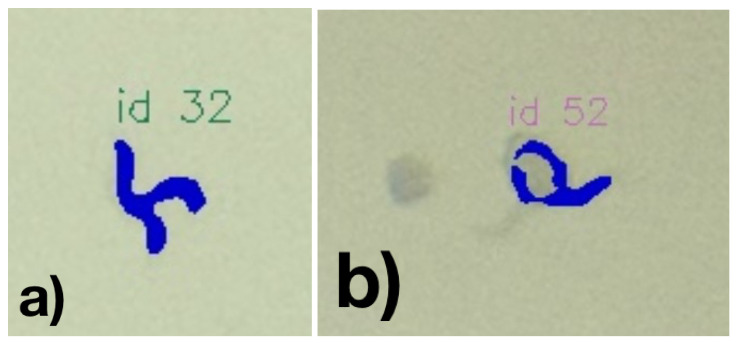
(**a**) Example of two crossing nematodes—detected as a single one and (**b**) an erroneous segmentation due to a worm being in a blob.

**Table 1 biosensors-11-00257-t001:** Evaluation Results as calculated via the manually annotated dataset; AP: Average Precision, IoU: Intersection over Union.

Approach	Precision	Recall	F1-Score	AP	Avg. Mask IoU
Correct classification of masks					
with any IoU	0.960	0.956	0.958	-	0.634
with ≥0.2 IoU	0.907	0.93	0.918	0.917	
with ≥0.3 IoU	0.912	0.934	0.923	0.923	
with ≥0.4 IoU	0.913	0.936	0.924	0.924	
with ≥0.5 IoU	0.896	0.917	0.906	0.902	0.789
with ≥0.6 IoU	0.864	0.886	0.875	0.864	
with ≥0.7 IoU	0.777	0.796	0.786	0.761	
with ≥0.8 IoU	0.499	0.510	0.504	0.412	
with ≥0.9 IoU	0.0081	0.0083	0.0082	0.0016	

## Data Availability

The dataset is available via: https://doi.org/10.5281/zenodo.5144802 (accessed on 26 July 2021).

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
