# Peer review of "Mask R-CNN Based C. Elegans Detection with a DIY Microscope"

_biosensors, 2021, doi:10.3390/bios11080257_

Round 1
Reviewer 1 Report
The authors present a study using Mask R-CNN based C. Elegans Detection with a DIY Microscope based on the Raspberry Pi platform.
The following questions arise and the following issues have to be addressed:
- The authors mention that measurements were done directly in a RAW format. Nevertheless, the usual output from the raspberry camera is a compressed JPEG file that has passed through all the stages of image processing to produce a high-quality image. How really were obtained raw files from lab-made instrumentation?.
- A similar issue is a precise control on exposure and aperture settings while taking the images. Have the pictures been recorded at fixed values of exposure time and aperture setting using the Raspberry Pi camera?. Authors claim the use of “visual inspection” for camera settings.
- ML-based segmentation and Mask R-CNN analysis are correct.
Reviewer 2 Report
This paper proposes a very interesting low cost image processing system to quantify C. elegans nematodes in Petri plates using an embedded system based on Raspberry Pi and Machine Learning algorithms.
Even in my opinion, this paper could be published in the present form, I would suggest some improvements to the authors:
- Figure 1 is illustrative of the overall scheme of the system, but a photograph of the hardware setup could help other researchers how this system could be mounted in a lab.
- Some image processing algorithms in preprocessing step are not clearly identified (e.g. bilateral filter uses a weighted average using gaussian kernel, but parameters like color distribution or space are not of general use. A mathematical formula or proper specific reference would help)
- I have a confusion about annotation. In line 208, authors stated that all images have been manually annotated, but it seems that the preprcessing steps are used to automate the annotation process. In my opinion, this need to be clarified, because it could be important for results.
- An IoU of 0.5 is used to determine a detection, in line 293 it seems that a study of model's sensitivity regarding its IoU has been done, but very few data of this model is presented (only basic metric scores). It will be nice to know the variation of those metrics with different values of IoU, as a higher value could be required when very accurate segmentation is required. It's not clear the difference of correct classification of masks and AP@0.5 IoU approaches in Table 1. Are both methods computed with reference to manually annotation dataset?
- Line 229 has a missed cite or a blank [].
